# Epigenetic activation of the *FLT3* gene by ZNF384 fusion confers a therapeutic susceptibility in acute lymphoblastic leukemia

Xujie Zhao [1,8], Ping Wang[2,8], Jonathan D. Diedrich[1], Brandon Smart[1], Noemi Reyes[1], Satoshi Yoshimura[1], Jingliao Zhang [1], Wentao Yang[1], Kelly Barnett[1], Beisi Xu [3], Zhenhua Li[1], Xin Huang [4], Jiyang Yu [4], Kristine Crews [1], Allen Eng Juh Yeoh [5], Marina Konopleva [6], Chia-Lin Wei [2], Ching-Hon Pui [7], Daniel Savic [1] & Jun J. Yang [1,7] ✉

*FLT3* is an attractive therapeutic target in acute lymphoblastic leukemia (ALL) but the mechanism for its activation in this cancer is incompletely understood. Profiling global gene expression in large ALL cohorts, we identify over-expression of *FLT3* in *ZNF384*-rearranged ALL, consistently across cases harboring different fusion partners with *ZNF384*. Mechanistically, we discover an intergenic enhancer element at the *FLT3* locus that is exclusively activated in *ZNF384*-rearranged ALL, with the enhancer-promoter looping directly mediated by the fusion protein. There is also a global enrichment of active enhancers within ZNF384 binding sites across the genome in *ZNF384*-rearranged ALL cells. Downregulation of *ZNF384* blunts *FLT3* activation and decreases ALL cell sensitivity to FLT3 inhibitor gilteritinib in vitro. In patient-derived xenograft models of *ZNF384*-rearranged ALL, gilteritinib exhibits significant anti-leukemia efficacy as a monotherapy in vivo. Collectively, our results provide insights into FLT3 regulation in ALL and point to potential genomics-guided targeted therapy for this patient population.

Treatment outcome of pediatric acute lymphoblastic leukemia (ALL) has improved significantly over the past decades, partially thanks to risk-adapted chemotherapy[1,2]. In particular, somatic genomics features of ALL can directly inform treatment stratification. For example, *ETV6-RUNX1* translocation is associated with a favorable prognosis even with low-intensity antimetabolite-based therapy whereas *BCR-ABL1* and *BCR-ABL1*-like have highly unfavorable prognosis despite intensive treatment including allogeneic hematopoietic cell. Recent advances in genomic profiling of ALL have led to the discovery of up to 22 molecular subtypes of B-ALL[3], many of which are defined by characteristic gene rearrangements, including *ZNF384*, *DUX4*, *BCL2*, *PAX5*, *NUTM1*,

and *HLF*. An improved understanding of ALL genomic landscape also enables the identification of new therapeutic targets and the exploration of novel therapy. For instance, patients with *KMT2A*-r ALL have generally poor treatment outcomes with conventional cytotoxic chemotherapy, and yet *KMT2A* itself is challenging to target therapeutically. Subsequent gene expression profiling studies identified *FLT3* as one of the most upregulated genes distinguishing *KMT2A*-r ALL from non-*KMT2A*-r ALL and acute myeloid leukemia (AML)[4], and its therapeutic potential has been extensively explored in laboratory models and patients[5–7]. However, the value of FLT3 inhibitor therapy for other ALL subtypes remains unclear.

[1]Department of Pharmacy and Pharmaceutical Sciences, St. Jude Children's Research Hospital, Memphis, TN, USA. [2]The Jackson Laboratory for Genomic Medicine, Farmington, CT, USA. [3]Center for Applied Bioinformatics, St. Jude Children's Research Hospital, Memphis, TN, USA. [4]Department of Computational Biology, St. Jude Children's Research Hospital, Memphis, TN, USA. [5]Department of Pediatrics, National University of Singapore, Singapore, Singapore. [6]Departments of Leukemia, The University of Texas MD Anderson Cancer Center, Houston, TX, USA. [7]Department of Oncology, St. Jude Children's Research Hospital, Memphis, TN, USA. [8]These authors contributed equally: Xujie Zhao, Ping Wang. ✉e-mail: jun.yang@stjude.org

FLT3 is a single transmembrane receptor belonging to the type III receptor tyrosine kinase family[8]. *FLT3* expression is mainly restricted to hematopoietic stem cells and early progenitors, suggesting its crucial role in early hematopoiesis[9–11]. Upon binding with its ligand (FL), FLT3 phosphorylates phosphoinositol-3-kinase (PI3K), growth factor receptor-bound protein (Grb2), and Src family tyrosine kinases, inducing proliferation of hematopoietic progenitors and promoting their survival. Aberrant activation of *FLT3* due to mutations or epigenetic deregulation can be oncogenic, contributing to the pathogenesis of ALL, AML, and chronic myeloid leukemia[12–14]. FLT3 internal tandem duplication (ITD) of the juxtamembrane domain and tyrosine kinase domain (TKD) mutations are among the most common genomic lesions in AML, although they are relatively rare in ALL. In *KMT2A*-rearrangement ALL, direct binding of KMT2A-fusion protein at the *FLT3* gene locus has been described, pointing to fusion protein-mediated transcriptional regulation of *FLT3*[15]. However, the underlying molecular mechanism of high *FLT3* expression in other ALL subtypes is still largely unknown.

*ZNF384*-rearrangement (*ZNF384*-r) was discovered recently as a recurrent genomic abnormality in ALL, with distinct biological and clinical features[16–19]. *ZNF384* gene encodes a C2H2-type zinc finger protein with potential transcription factor activity[17,20]. In B-ALL, *ZNF384* can be fused with 22 partner genes including *EP300*, *TCF3*, *TAF15*, and *CREBBP*, but they give rise to a common gene expression signature that defines this subtype. Even though the 5-year overall survival is relatively high for *ZNF384*-r ALL, about 30% of patients in this group are classified as high-risk group[21]. Recently, high expression of *FLT3* was described in an adult ALL patient with the *EP300-ZNF384* fusion, with an impressive clinical response to FLT3 inhibitor therapy[22], suggesting leukemia dependency of FLT3 signaling in this context.

In this study, we elucidated the molecular mechanisms of ZNF384-mediated *FLT3* activation in *ZNF384*-r ALL and identified *cis*-regulatory elements responsible for transcriptional activation of *FLT3*. We further evaluated therapeutic potential of gilteritinib[23,24], an FLT3 inhibitor, in *ZNF384*-r ALL in vitro and in vivo, pointing to potential genomics-guided targeted therapy for this patient population.

## Results

### Subtype-specific activation of *FLT3* in childhood ALL

We first examined *FLT3* expression in different ALL molecular subtypes using published RNA-seq dataset of 1988 B-ALL patients (Cohort 1) (mostly of European descent), including 1,610 children and 378 adults[3]. As shown in Fig. 1a, *FLT3* expression was noted in most of ALL subtypes (with the exception of *NUTM1*, *BCL2/MYC*, and *IKZF1 N159Y*), although there was a great degree of heterogeneity within each subtype. The highest *FLT3* expression was observed in 49 cases with *ZNF384* rearrangements and 136 subjects with *KMT2A* fusion genes. In particular, *FLT3* was consistently overexpressed in ALL cases harboring *ZNF384* fusions with a variety of partner genes, e.g., *EP300*, *TCF3*, *TAF15*, *ARID1B*, *CLTC*, *CREBBP*, *EWSR1*, *NIPBL*, and *SMARCA2* (Fig. 1b). Similarly, this pattern of *FLT3* overexpression was also observed in a pediatric ALL cohort of Asian descent (Cohort 2) (Fig. 1c), again with consistent activation across *ZNF384* fusions (Fig. 1d)[25]. *FLT3* mutation was uncommon in *ZNF384*-rearranged ALL, identified in 5 of 66 cases (7.6%) from these two cohorts (Fig. 1e).

### ZNF384 fusion transcriptionally activated *FLT3* and affected local chromatin looping

Given the known function of ZNF384 as a transcription factor, we hypothesized that *FLT3* is transcriptionally activated by ZNF384 fusion protein in ALL. To explore this, we first performed a CUT&RUN assay in a patient-derived xenograft (PDX) ALL sample that harbors the *TCF3-ZNF384* fusion (sample ID: TCZ). In this assay, we profiled ZNF384

binding as well as H3K27ac, and H3K4me3 modification across the genome. As shown in Fig. 2a, b, a total of 10,470 ZNF384 binding peaks were identified, with 4820 peaks in promoter regions (i.e., ±2 Kb of transcription start site) and 6650 in putative enhancer regions (i.e., intronic or intergenic). Of ZNF384 binding sites, 6750 (65.5%) and 4602 (34.5%) overlapped with H3K27ac and H3K4me3 marks, respectively, pointing to broad transcription activator activity of ZNF384 fusion protein. Additionally, we also profiled ZNF384 binding in an *EP300-ZNF384* ALL cell line JIH5 and observed similar results (Supplementary Fig. 1).

Importantly, a prominent ZNF384 binding site was identified in the 5′ distal region of the *FLT3* gene, 25 kb from *FLT3* promoter (Fig. 2c). This region was also marked with sharp H3K27ac and H3K4me3 modifications, characteristic of an active enhancer. Using the luciferase assay in JIH5 cells, this 500 bp segment (hg38, chr13:28,124,365–28,124,868) increased transcription activity by 12.7-fold compared to vector control (Fig. 2d). Furthermore, we also examined chromatin accessibility at the *FLT3* locus in primary ALL blasts of diverse subtypes, using ATAC-seq. Again, the ZNF384 binding site was precisely located within an open chromatin region specific to cases with *ZNF384* fusion (sample ID: EPZ 1 and EPZ 2) (Fig. 2c middle panel), in agreement with its enhancer activity. By contrast, this enhancer region was completely void of ATAC-seq signals in ALL samples of other molecular subtypes (*TCF3-PBX1*, *DUX4* rearrangement, *ETV6-RUNX1*, and *KMT2A-AF4*), suggesting a causal effect of ZNF384 fusion protein (Fig. 2c bottom panel). Notably, this z-FLT3 enhancer was not observed in other cancer cell lines in the ENCODE dataset (Supplementary Fig. 2).

Because this ZNF384 binding site is distal to the transcription start site of *FLT3*, we hypothesized that long-distance chromatin looping is needed to tether together this enhancer and the *FLT3* promoter. To test this hypothesis, we performed ChIA-PET assay[26] in JIH5 cells as well as PDX-derived ALL blasts with *TCF3-ZNF384* fusion. As shown in Fig. 3, there was prominent DNA looping involving *FLT3* promoter and enhancer, mediated by both CTCF and RNA polymerase II (RNAP II) in both JIH5 cells and PDX-derived ALL blasts. In comparison, this interaction was absent in B lymphoblastoid cell line GM12878, which lacks *ZNF384* fusion. To explore the involvement of ZNF384 fusion in the complex looping architecture, we also performed ChIA-PET using ZNF384 antibody and directly confirmed that this enhancer-promoter looping was mediated by the fusion protein in PDX-derived ALL blasts and in JIH5 cells. Genome-wide chromatin interactions mediated by ZNF384 fusion protein were summarized in Supplementary Data 1 and 2.

### ALL cells with high *FLT3* expression were preferentially sensitive to FLT3 Inhibition

Eleven ALL cell lines (Nalm6, SEM, 697, UOCB-1, REH, JIH5, RPMI8402, CEM, Jurkat, DND41, and MOLT4) representing *DUX4*, *KMT2A-r*, *TCF3-PBX1*, *TCF3-HLF*, *ETV6-RUNX1*, *ZNF384-r*, and T-ALL subtypes were subjected to drug-sensitivity assay in vitro, using FLT3 inhibitor gilteritinib. After 72 hours of exposure, LC50 was calculated to quantify drug sensitivity (Fig. 4a). SEM (LC50: 44 nM) and JIH5 (LC50: 3.9 nM) cells were the most sensitive to gilteritinib, consistent with the overexpression of *FLT3* in *KMT2A*-r and *ZNF384*-r ALL (Fig. 1a). In fact, gilteritinib LC50 was negatively correlated with *FLT3* mRNA expression ($r = -0.9$, $P = 0.0002$) and protein expression on the cell surface ($r = -0.8364$, $P = 0.0013$) across 11 ALL cell lines (Supplementary Fig. 3). To further confirm the selective sensitivity of *ZNF384*-r ALL to gilteritinib, we performed ex vivo drug-sensitivity assay in a panel of 47 primary ALL cases including three with *EP300-ZNF384* fusion as well as those with *ETV6-RUNX1*, *DUX4*-r, *BCR-ABL1*, hyperdiploidy, and T-ALL. Primary ALL samples with *ZNF384* fusion showed superior drug sensitivity than those of other subtypes (Fig. 4b). To directly prove the regulation of *FLT3* by EP300-ZNF384, we knocked down fusion gene

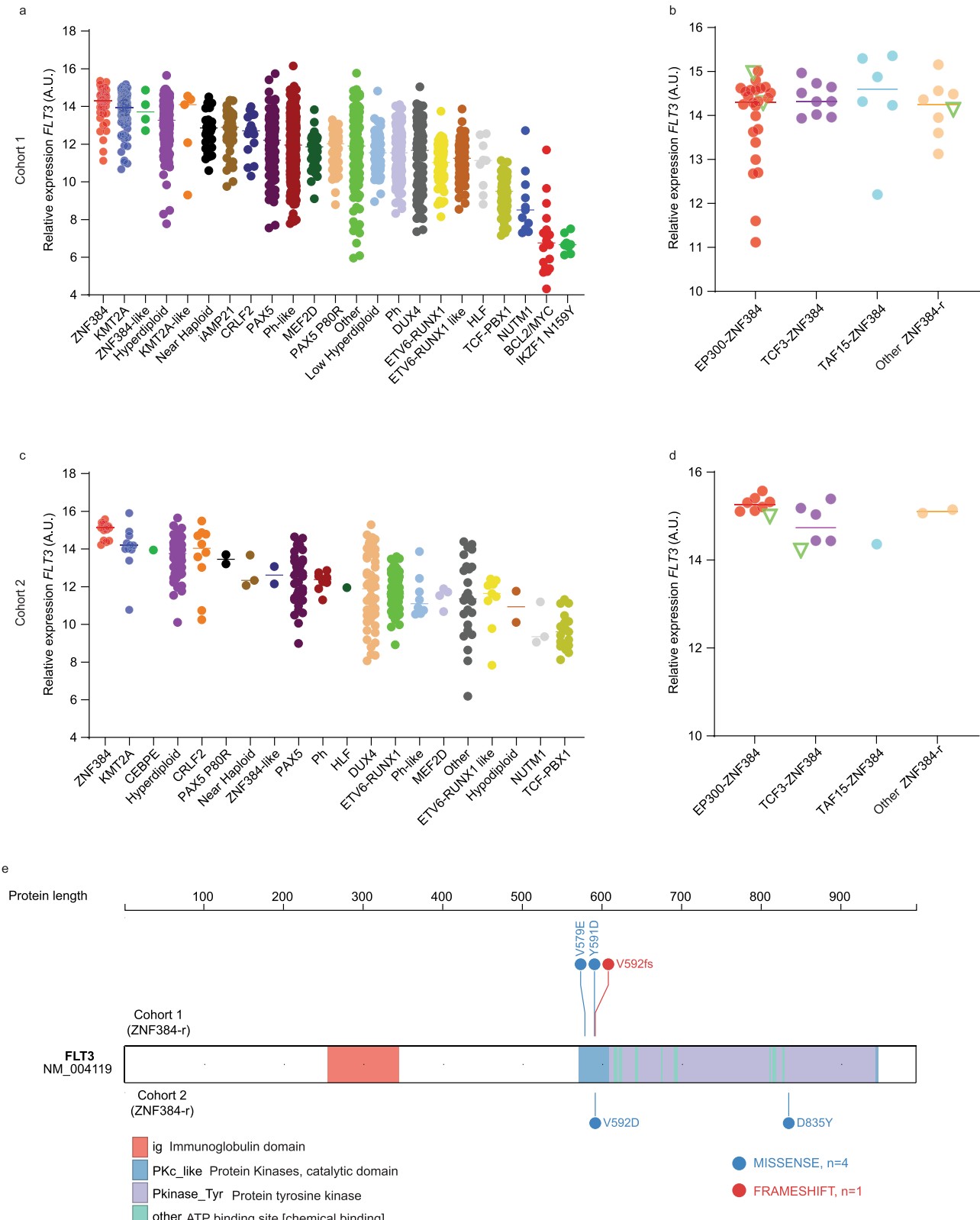

**Fig. 1 | *FLT3* expression by subtype in B-ALL patients of diverse ancestries.**
**a** *FLT3* expression across different B-ALL subtypes in the US cohort of 1988 children
and adults (Cohort 1), with the highest expression of *FLT3* in *ZNF384*-r (*ZNF384*-
rearrangement) ALL (*n* = 1988). **b** *FLT3* expression in ALL cases (*n* = 49) with various
*ZNF384* fusions in cohort 1. Subgroups were defined by fusion partners of *ZNF384*.
Other: fusion partners include *ARID1B, CLTC, CREBBP, EWSR1, NIPBL*, and *SMARCA2*.
**c** *FLT3* expression across different B-ALL subtypes in cohort 2 (*n* = 377), with the
highest expression of *FLT3* in *ZNF384*-r ALL. **d** *FLT3* expression across *ZNF384*
fusions (*n* = 17) in the Asian cohort (Cohort 2). Subgroups were defined by fusion
partners of *ZNF384*. Other: fusion partners include *USP25* and *CREBBP*. **e** *FLT3*
mutations in ZNF383-r ALL from each ALL cohort. *A.U.* arbitrary units. Center lines
indicate median values of *FLT3* expression. Source data are provided as a Source
Data file.

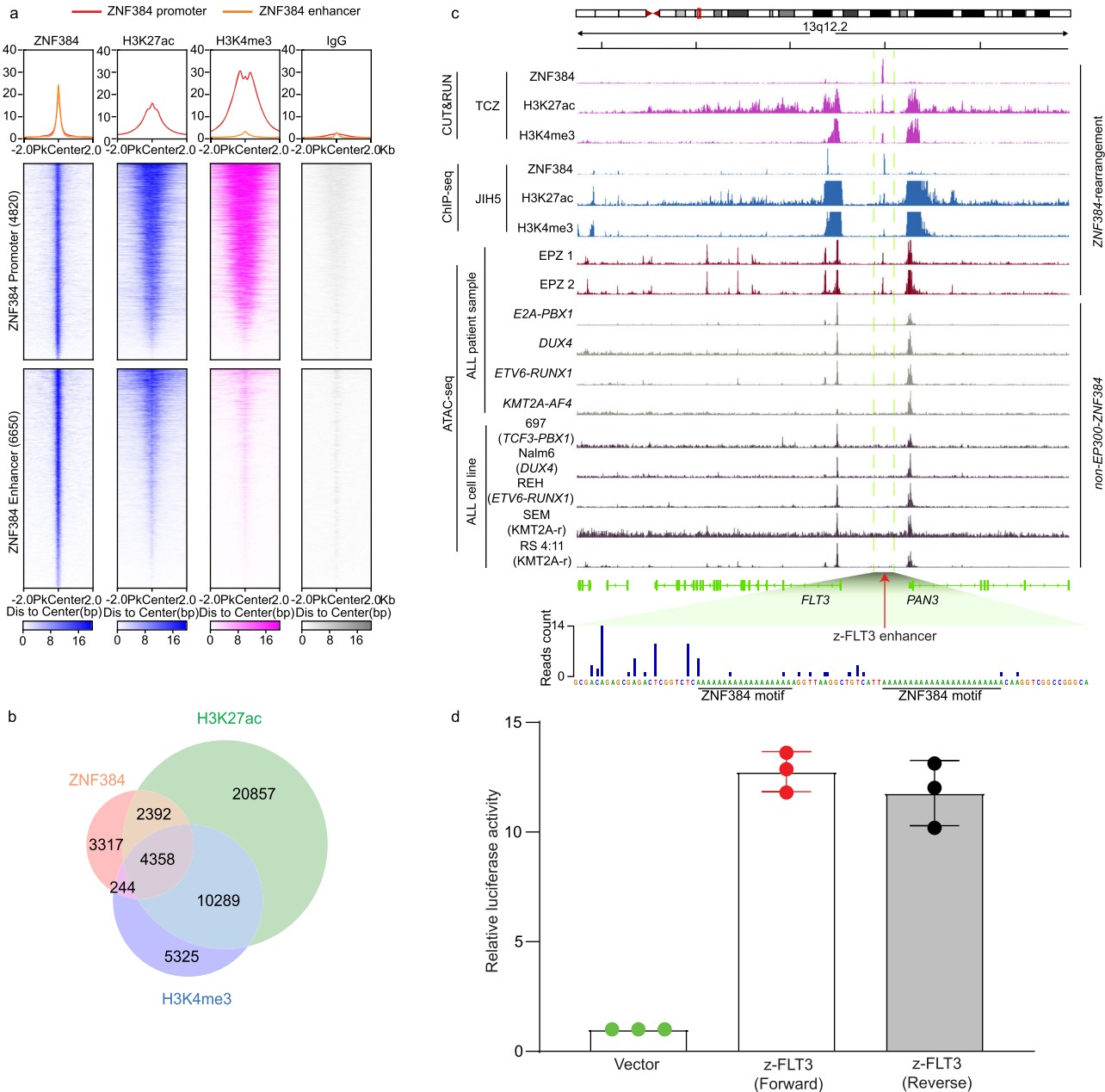

**Fig. 2 | ZNF384 fusion-specific activation of a distal enhancer at the *FLT3* locus in ALL. a, b** ZNF384 CUT&RUN (Cleavage Under Targets & Release Using Nuclease) assay using a *TCF3-ZNF384* ALL PDX sample (ID TCZ). In total, 10,470 peaks were identified, with 4820 in promoter regions (±2 kb from TSS) and the other 6650 in enhancers. 6750 peaks overlapped with H3K27ac peaks, while 4602 peaks overlapped with H3K4me3. **c** CUT&RUN and ATAC-seq (Assay for Transposase-Accessible Chromatin using sequencing) of ALL cell lines and primary ALL blast samples with or without *ZNF384*-rearrangements. Each track represents a type of assay in a given sample as indicated on the left. As shown in the CUT&RUN tracks, a unique ZNF384 binding site was observed 25 kb upstream of *FLT3* with prominent H3K27ac and H3K4me3 marks. This peak also overlapped with an open chromatin region identified by ATAC-seq in *EP300-ZNF384* ALL primary samples (*n* = 2).

In contrast, this region was void of ATAC-seq signals in ALL cell lines or ALL primary samples of other subtypes. Bottom panel: two ZNF384 binding motifs (AAAAAAAA) were identified by footprint analysis using the ZNF384 CUT&RUN data derived from *TCF3-ZNF384* ALL PDX cells. **d** Enhancer activity of z-FLT3 enhancer was confirmed by luciferase assay. A 500 bp segment (hg38, chr13:28,124,365–28,124,868) covered by this z-FLT3 enhancer increased transcription activity by 12.7-fold compared to vector control in JIH5 cells. Enhancer activity was normalized to empty vector control. Data are shown as mean values ± SEM of three biological replicates (center of the error bar) and the results are representative of three independent experiments. Source data are provided as a Source Data file.

expression using the CRISPR/Cas9-mediated editing in JIH5 cells (Fig. 4c, Supplementary Fig. 4a), which led to 48% decrease of FLT3 expression (Fig. 4c, Supplementary Fig. 4b). Downregulating *EPZ* fusion expression led to a modest reduction in cell viability but without affecting cell cycle in JIH5 cells (Supplementary Fig. 4c, d). Similarly, knocking down *FLT3* resulted in decreased viability of JIH5 cells but not ALLs of other subtypes (i.e., 697 cells [*TCF3-PBX1* fusion positive] or

REH cells [*ETV6-RUNX1* fusion positive], Supplementary Fig. 5). Of note, downregulation of the fusion gene did not affect the expression of *PAN3* gene, a nearby gene of *FLT3* (Supplementary Fig. 6a). Consistently, downregulation of *EPZ* also resulted in ~100-fold decrease in JIH5 cell sensitivity to gilteritinib (*P* = 0.006) (Fig. 4d). Taken together, these results indicated that ZNF384-driven *FLT3* activation conferred ALL sensitivity to FLT3 inhibitor in vitro.

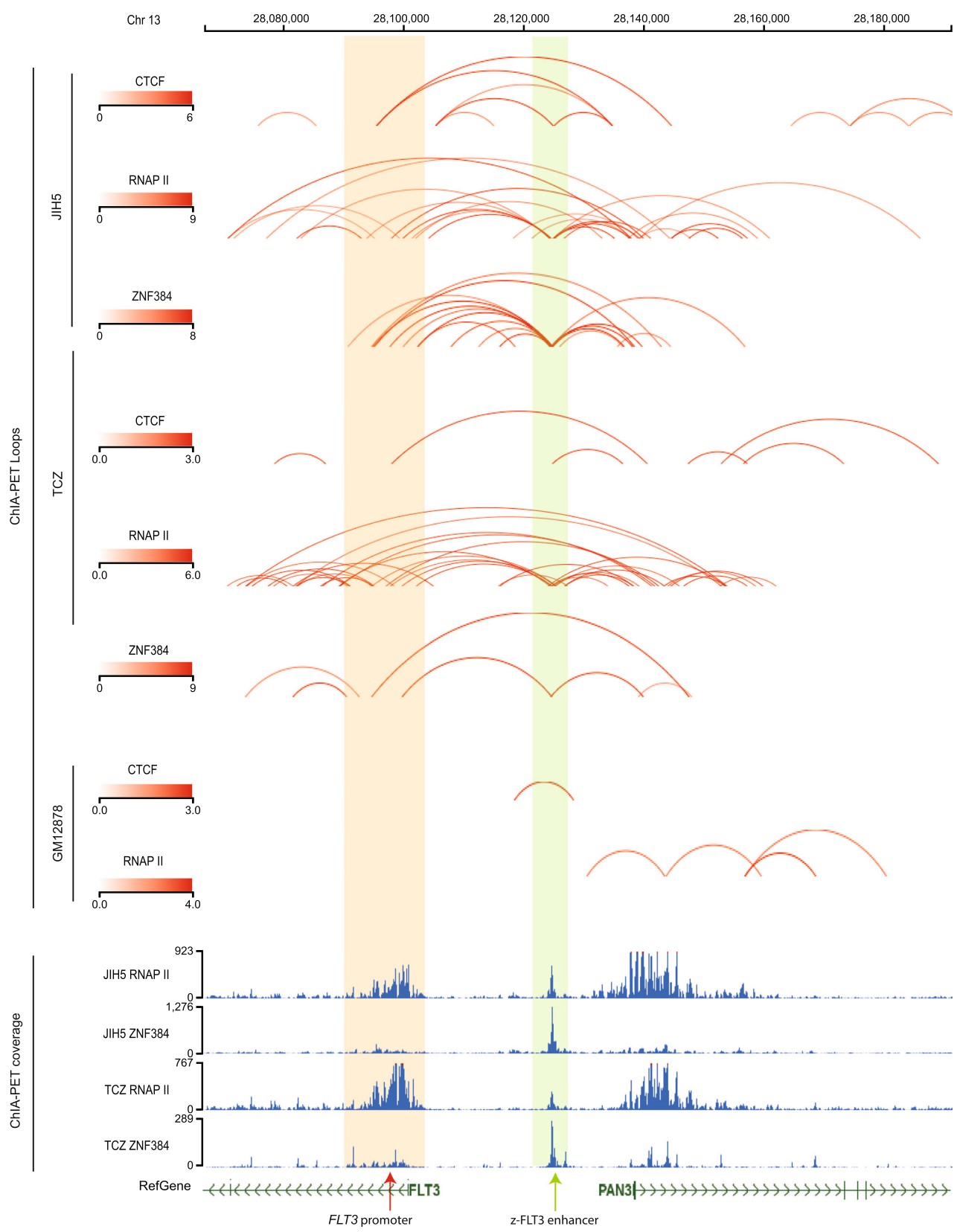

To further test the therapeutic effects of gilteritinib in vivo, we evaluated its anti-leukemic efficacy using two xenograft models of *ZNF384*-r ALL (TCZ and JIH5). Mice were treated with gilteritinib (10 mg/kg) or vehicle orally three days after injection. Human leukemia blast percentage in peripheral blood was monitored weekly using flow cytometry. As shown in Fig. 4e–h, gilteritinib treatment led to a marked delay in leukemia progression compared to vehicle control (panel e: $P < 0.005$, panel g: $P < 0.005$) and also longer leukemia-free survival (panel f: $P = 0.0027$, panel h: $P = 0.0026$), despite variation in the in vivo growth rate of these two *ZNF384*-r ALL models. Taken together, these results pointed to gilteritinib as a potential therapeutic strategy for patients with *ZNF384*-r ALL.

**Fig. 3 | 3D chromatin interaction at 5′-UTR of *FLT3* in *ZNF384*-r ALL.** ChIA-PET (chromatin interaction analysis with paired-end tags) (CTCF and RNAP II) was performed in both JIH5 cells and an ALL PDX sample harboring the *TCF3-ZNF384* fusion. As shown in the top four panels, DNA looping mediated by CTCF and RNAP II was detected between ZNF384-binding site ("z-FLT3 enhancer") and *FLT3* promoter, indicated by shaded box. In addition, we also performed ZNF384 ChIA-PET in both JIH5 cells and *TCF3-ZNF384* ALL PDX sample to map ZNF384-mediated chromatin interactions, in comparison to lymphoblastoid cell line GM12878 which does not harbor *ZNF384* fusion gene. Red curves indicated chromatin interactions called using ChIA-PIPE (paired-end tag reads ≥2) and were highlighted in red in Supplementary Data 1 and 2. Additionally, the bottom panels show protein binding sites (ZNF384 and RNAP II) identified from inter-ligation and self-ligation reads of ChIA-PET using MACS2 calling algorithm. Overlap of ZNF384 binding sites and chromatin looping anchors suggests enhancer-promoter interaction specifically mediated by ZNF384 (indicated by green and red arrows).

## Discussion

In this study, we identified high expression of *FLT3* as a transcriptomic feature of *ZNF384*-r ALL. By exploring the genome-wide binding pattern of ZNF384 fusion protein, we identified a ZNF384-driven enhancer at a distal open chromatin region of *FLT3* and elucidated molecular details of the enhancer-promoter interaction mediating *FLT3* activation. Our data demonstrated that *FLT3* transcription can be activated specifically by ZNF384 fusion protein in ALL, therefore giving rise to a potential therapeutic opportunity for this group of patients.

While *ZNF384* is widely expressed in almost all subtypes of ALL, *FLT3* is rarely overexpressed in cases without *ZNF384* fusion. Therefore, it is likely that ZNF384 fusion proteins exhibit gain-of-activity phenotype or acquire functions absent in wildtype ZNF384. This is in line with our previous report that EP300-ZNF384 fusion showed a significantly greater transcription factor activity than wildtype ZNF384[17]. It is also possible that this chimeric protein can uniquely recruit other transcription factors to co-activate target genes that are only modestly activated by wildtype ZNF384. Interestingly, when we examined the ZNF384-binding enhancer at the *FLT3* locus, we indeed observed binding motif of a variety of transcription factors (e.g., ETV1, RUNX1, NF-Y) in close proximity of that of ZNF384. However, this needs to be verified by pull-down assays using antibodies specific to wildtype ZNF384 or the fusion protein, respectively. If these reagents were to become available in the future, one should also consider ChIP-seq to identify differences in genome-wide binding pattern of wildtype vs fusion ZNF384 to fully understand epigenomic program driven by each.

It is important to note that *FLT3* overexpression is also observed in ALL cases without *KMT2A* or *ZNF384* rearrangements (Fig. 1a). For example, microdeletion at this locus in hyperdiploid ALL creates enhancers that directly activate *FLT3* transcription[27]. Therefore, it is plausible that there are a multitude of putative *cis*-regulatory elements around the *FLT3* gene and they can be activated by a variety of mechanisms, e.g., genomic alteration, presence of fusion genes, and upregulation of other transcription factors as a result of the sentinel genomic abnormalities. Therefore, there is a need to systematically identify these regulatory DNA sequences at this locus, to fully understand the epigenomic mechanisms of *FLT3* expression in ALL. Interestingly, *FLT3* expression seems to be linked to B-cell differentiation stages of ALL, higher in immature ALLs (*KMT2A*-r and *ZNF384*-r) and lower in the more mature subtypes (*ETV6-RUNX1*, *TCF3-PBX1*, and *MEF2D*-r) (Supplementary Fig. 7). By contrast, the nearby gene at this locus, *PAN3*, was ubiquitously expressed across different ALL subtypes (Supplementary Fig. 6b, c).

As a tyrosine kinase, FLT3 represents an attractive therapeutic target, and the therapeutic effects of inhibiting FLT3 have been explored in leukemia, particularly in AML with FLT3 ITD[7,22,28–33]. In ALL, clinical trials by COG and others evaluated quizartinib and lestaurtinib[34,35], with mixed results. For example, the COG AALL0631 trial tested the effects of lestaurtinib in infants with *KMT2A*-r ALL, and unfortunately the addition of this FLT3 inhibitor failed to improve event-free or overall survival[34]. Interestingly, these patients were also tested for leukemia blast sensitivity to lestaurtinib in vitro which was highly correlated with in vivo response[34]. It is unclear whether *FLT3* overexpression alone would confer response to FLT3 inhibitors in *KMT2A*-r ALL or is merely a consequence of other genomic events and thus not essential for leukemia cell survival. But at least in *ZNF384*-r ALL, there seems to be a dependency of FLT3 signaling although its exact contribution to leukemogenesis and leukemia maintenance remains unclear.

In summary, our findings point to a fusion protein-driven epigenetic mechanism of *FLT3* activation in ALL and expand the armory of genomics-guided targeted therapy for this leukemia.

## Methods

### ALL genomics data

Two previously published ALL RNA-seq datasets were included in this study: (1) 1988 cases of pediatric, adolescent, and young adult ALL enrolled at St. Jude Children's Research Hospital (St. Jude), Children's Oncology Group (COG), ECOG-ACRIN Cancer Research Group (ECOG-ACRIN), the Alliance for Clinical Trials in Oncology (Cancer and Leukemia Group B), M.D. Anderson Cancer Center (MDACC), University of Toronto, Northern Italian Leukemia Group (NILG), Southwestern Oncology Group (SWOG), Medical Research Council UK, and City of Hope treatment protocols (EGAS00001003266)[3]; (2) 377 pediatric ALL cases from the Ma-Spore ALL 2003 (MS2003) and Ma-Spore ALL 2010 (MS2010) studies (EGAS00001001858, EGAS00001003726, and EGAS00001004532)[25]. This project was approved by institutional review boards at St. Jude Children's Research Hospital, and informed consent was obtained from parents, guardians, or patients, as appropriate. No compensation was provided to patients or parents for participating this research.

### Cell Culture

Nalm6 (CRL-3273), REH (CRL-8286), MOLT4 (CRL-1582), CEM (CRL-2265), and Jurkat (TIB-152) cell lines were purchased from ATCC, and JIH5 (ACC 788), SEM (ACC 546), 697 (ACC 42), RPMI8402 (ACC 290), and DND41 (ACC 525) cell lines were purchased from DSMZ, and UOCB-1 (a kind gift from Dr. William Evans at St. Jude Children's Research Hospital, detailed information of this cell line is available at Cellosaurus database with accession number: CVCL_A296). All cell lines (except for JIH5) were grown in RPMI 1640 medium (GIBCO, #11875093) supplemented with 10% heat-inactivated fetal bovine serum and two mM L-glutamine, at 37 °C in 5% $CO_2$. JIH5 cells were grown in IMEM medium (GIBCO, #A1048901) supplemented with 20% heat-inactivated fetal bovine serum and two mM L-glutamine, at 37 °C in 5% $CO_2$.

### CRISPR knockout of EP300-ZNF384 fusion gene

CRISPR/Cas9 was used to knock out the *EP300-ZNF384* (*EPZ*) fusion gene in JIH5 cells. Specific sgRNAs targeting *EP300* part of the *EPZ* fusion gene were designed using the online software CHOP–CHOP (https://chopchop.cbu.uib.no/). The paired sgRNA oligos were subcloned into px458 vector (Addgene, #48138) and electroporated into five million JIH5 cells using the Nucleofector-2b device with Kit R (Lonza, #VVCA-1001) and program T-016. The mRNA expression levels of *EPZ* and *FLT3* were confirmed by quantitative real-time PCR (qRT-PCR) following the procedures described below. Total RNA was collected by using the RNeasy Miniprep Kit (QIAGEN, #74106). cDNA synthesis was performed by using SuperScript IV One-Step RT-PCR

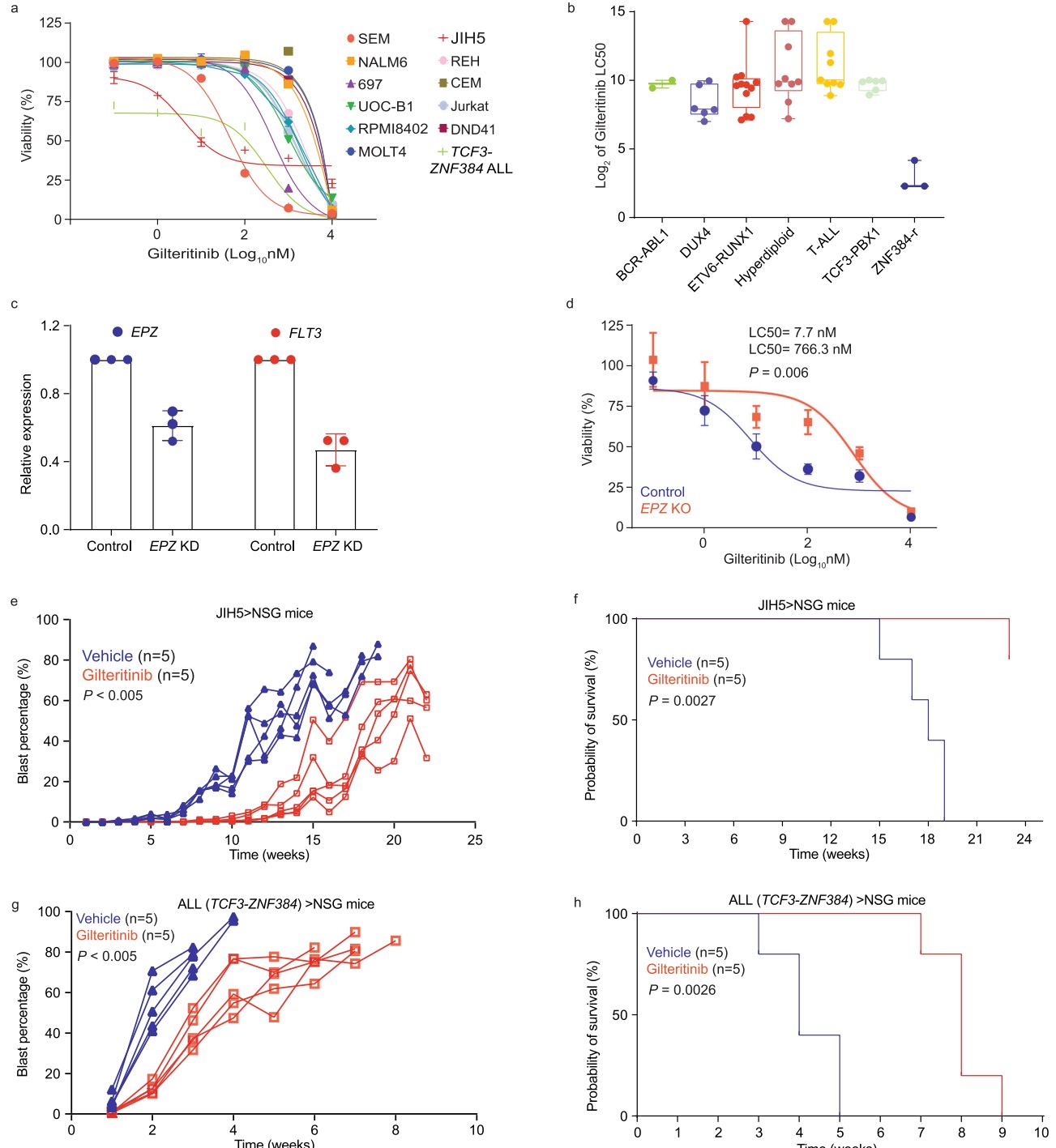

**Fig. 4 | Anti-leukemia effect of gilteritinib in vitro and in vivo. a** In vitro sensitivity of a panel of ALL cell lines as well as PDX-derived ALL cell with *TCF3-ZNF384* fusion to gilteritinib was determined using MTT assay. Data are shown as mean % viability relative to vehicle ± SEM of three biological replicates (center of the error bar) and results are representative of three independent experiments. **b** Ex vivo sensitivity of a panel of 47 primary ALL cases including three with *EP300-ZNF384* fusion as well as those with *ETV6-RUNX1*, *DUX4*-r, *BCR-ABL1*, hyperdiploidy, and T-ALL. Box plots show summary of data in terms of minimum, maximum, median, and first and third quartiles. Each data point represents two technical replicate. **c**, **d** Expression of *EPZ* (*EP300-ZNF384*) fusion gene was downregulated by CRISPR Cas9 editing in JIH5 cells, which led to decreased expression of *FLT3* and drug sensitivity to gilteritinib. Data are shown as mean values ± SEM of three biological

replicates (center of the error bar) and results are representative of three independent experiments. *P* values (*P* = 0.006) were estimated using two-sided *t* test. **e**−**h** Gilteritinib efficacy was evaluated in vivo using xenograft models. **e**, **f** Show leukemia progression, and survival in mice transplanted with JIH5 cells. **g**, **h** Describe results of mice transplanted with PDX-derived ALL cells with *TCF3-ZNF384* fusion. Gilteritinib was given daily at a dosage of 10 mg/kg, and therapy started three days following leukemia engraftment. Leukemia burden was monitored weekly, and *P* values (*P* < 0.005) were estimated using a two-sided analysis of deviance based on mixed effect models with cubic splines. Leukemia-free survival was plotted as Kaplan−Meier curves and *P* values were estimated using two-sided log-rank test (*P* = 0.0027 for **f**, *P* = 0.0026 for **h**). Source data are provided as a Source Data file.

System (Thermo Fisher, #12594100). Real-time PCR was performed by using the FAST SYBR Green Master Mix (Applied Biosystems, #4385612) following the manufacturer's instructions. Relative gene expression was determined by using the ΔΔ-CT method[36]. Sequences of sgRNA oligos and primers were described in Supplementary Tables 1–2 and positions of sgRNA and primers were shown in Supplementary Fig. 8.

## *FLT3* knock down

*FLT3* expression was knocked down in JIH5 cells using siRNA (Santa Cruz, #sc-29320). In REH, 697, and SEM cells, *FLT3* was downregulated using CRISPRi/dCas9-KRAB. REH, 697, and SEM cells with stable expression of dCas9-KRAB were established using lenti-dCas9-KRAB-last plasmid (Addgene, #89567). sgRNA (GACCGCAAATTCCCTCG-GAC) targeting the TSS region of *FLT3* gene was designed and cloned into LentiGuide-Puro (Addgene, #52963) plasmid. dCas9-KRAB cells were transduced with sgRNA lentivirus for two days and selected with puromycin (2 μg/mL) for 2–3 days. All the high-titer lentivirus were generated with Lenti-X cells (Takara, #632180).

## ATAC-seq

ATAC-seq was performed following the Fast-ATAC protocol[37]. Briefly, 10,000 of cells were treated with transposition mix: (1) 22 μL nuclease-free water; (2) 25 μL 2× TD Buffer; (3) 2.5 μL TDE1 transposase; (4) 0.5 μL 1% Digitonin (added right before removing supernatant from cell pellets), followed by incubation at 37 °C for 30 minutes. DNA was purified by using the MinElute Reaction Cleanup kit (Qiagen, #28204) and amplified for five cycles with the following mix: 10 μL transposed DNA, 2.2 μL nuclease-free water; 6.25 μL 10 μM barcoded Nextera primer-F; 6.25 μL 10 μM barcoded Nextera primer-R; 0.3 μL 100× SYBR Green I (Invitrogen, #S-7563); 25 μL NEBNext® High-Fidelity 2× PCR Master Mix (NEB, #M0541). PCR products were subsequently re-amplified for six additional cycles and subjected to purification using SPRIselect beads (Beckman Coulter, #B23317). Paired-end sequencing (2 × 100 bp) was conducted using the Illumina HISeq 2500 platform. Paired-end reads were applied on cutadapt (version 1.18)[38] for adaptor trimming and then mapped to the human hg19 genome reference by Bowtie2 (version 2.2.9)[39]. Peak calling was performed by MACS2[40] with default parameters on each sample. Peaks from all samples were merged by bedtools (version 2.25.0)[41] to retain non-overlapped regions. These regions were then used for identifying differentially enriched peaks, which was performed by ABSSeq under the aFold model[42] with read count from HTSeq[43]. The cutoff of adjusted *P* values <0.05 and log$_2$ fold change ≥ 2 was used to define high-confidence ATAC-seq peaks. Enriched regions were mapped to the nearest gene in human hg19 by Homer[44].

## Western blotting

Total protein was extracted with RIPA buffer (Thermo Scientific, #89901) supplemented with protease and phosphatase inhibitor cocktail (Thermo Scientific, #78440). Protein lysates were incubated on ice with gentle shaking for 15 minutes before being centrifuged at 4 °C/20,000 × *g* for 15 minutes. Supernatants were mixed with an equal volume of 2-mercaptoethanol (Bio-Rad, #1610710) supplemented 2× laemmli sample buffer (Bio-Rad, #1610737). Protein samples were run on precast 4–15% Tris-glycine Mini-PROTEAN TGX gels (Bio-Rad, #4561086) and transferred to an Immobilon-FL PVDF membranes (Millipore, #IPFL00010) at a constant 100 V for one hour. After incubation with Intercept® (TBS) blocking buffer (LI-COR, #927-60001) for one hour at room temperature, the membrane was incubated with antibodies against Actin (CST, #4970 S, dilution ratio: 1:2000), ZNF384 (Abclonal, # A15964, dilution ratio: 1:500) overnight at 4 °C with gentle shaking. Membranes were washed with TBST buffer three times for 30 min and incubated with the IRDye® 800CW goat anti-rabbit (LI-COR, #926-32211, dilution ratio: 1:2000) for 2 hours at room

temperature. Excessive antibodies were washed with TBST buffer and the membranes were exposed with the LI-COR Odyssey XF Imager (LI-COR).

## Flow cytometry to quantify FLT3 expression, cell cycle distribution, and apoptosis

Cells were stained for APC anti-human FLT3 antibody (Invitrogen, #17-1357-42, dilution ratio: 1:100) followed by flow cytometry analysis using BD LSR Fortessa (BD Biosciences) and data analysis was performed using FlowJo software (Tree Star). For cell cycle analysis, cells were fixed with the BD Cytofix/Cytoperm kit (BD Biosciences, #554714) after cell surface staining followed by DAPI staining. Staining of apoptotic cells was done by cell surface staining, followed by labeling with Annexin V (BD Biosciences, #556420) and DAPI in Annexin V staining buffer (BD Biosciences, #556454). Gating strategy is shown in Supplementary Fig. 9.

## Luciferase reporter assay

According to ZNF384 CUT&RUN results in JIH5 cells, a 500 bp region containing ZNF384 binding site (hg38, chr13:28,124,365-28,124,868) (forward) and a 500 bp fragment with reverse sequence (reverse) were cloned into a pGL4.23 (luc2/miniP) luciferase vector (Promega, #EB8411). For luciferase reporter assay, 5 × 10$^6$ JIH5 cells cultured in 12-well plate were transiently transfected with 6 μg of pGL4.23 construct and 1 μg of pGL-TK (Renilla luciferase) (Promega, #E6921) using Nucleofector-2b device (Lonza) with Kit R (#VCA-1001) and program T-016. Firefly luciferase activity was measured 24 hours post-transfection and normalized to Renilla luciferase activity. Relative luciferase activity indicates the ratio normalized to the value from empty pGL4.23 vector. All experiments were performed in triplicate and repeated three times.

## CUT&RUN and data analysis

CUT&RUN was performed using the CUT&RUN assay kit (CST, #86652). Cells were first coated with magnetic beads, followed by permeabilization with digitonin. Antibodies against ZNF384 (Abclonal, #A15964, 4 μg), H3K27ac (CST, #8173 S, 4 μg), or H3K4me3 (CST, #9751, 4 μg) were then incubated with permeabilized cells. pAG-MNase enzyme was then added to digest exposed genomic DNA. Finally, the DNA complex was released into the supernatant and was purified for next-generation sequencing.

Paired-end reads of 100 bp were mapped human genome hg19(GRCh37-lite)by BWA (version 0.7.12-r1039, default parameter)[45] after trimming for adapters by fastp(version 0.20.0, paired-end mode, parameters "--trim_poly_x --cut_by_quality5 --cut_by_quality3 --cut_mean_quality 15 --length_required 20 --low_complexity_filter --complexity_threshold 30 --detect_adapter_for_pe")[38], duplicated reads were then marked with biobambam2(version v2.0.87)[46] and only non-duplicated reads have been kept by samtools (parameter "-q 1 -F 1804" version 1.2)[47]. All samples have more than 5 M fragments as suggested by Cut&Run protocols paper[48]. We then generated bigwig files using the center 80 bp of fragments smaller than 2000 bp and normalized to 10 M fragments. Peaks were called by MACS2 (version 2.1.1.20160309, "--extsize 200 --nomodel --keep-dup all")[40]. For Venn Diagram, we merge peaks from ZNF384, H3K4me3, and H3K27ac to get the reference peaks set then count whether a reference peak overlaps ZNF384, H3K4me3, and H3K27ac. We then separated ZNF384 peaks into promoter (TSS ± 2 kb) or enhancer based on Gencode annotation database (v31lift37)[49] and deeptools (v2.5.7)[50] were used to generate heatmap.

## ChIA-PET

In situ ChIA-PET libraries with antibodies against RNAPII (BioLegend, #664906, 20 μg), CTCF (Abclonal, #A1133, 20 μg), and ZNF384 (Abclonal, #A15964, 20 μg) were generated using ~10 million input

cells from JIH5 and *TCF3-ZNF384* ALL PDX cells, following the in situ ChIA-PET protocol[51], with the following steps: (1) cells were crosslinked by standard formaldehyde treatment, and the pellets were then snap-frozen in liquid nitrogen; (2) the crosslinked cells were lysed to release the chromatin–DNA complexes followed by fragmentation to an average size of 300 base pairs. (3) The sonicated chromatin–DNA complexes were incubated with the antibody-coated magnetic Protein G beads (Covance, #8WG16). The ChIA-PET libraries were sequenced by 2 × 150 bp paired-end reads using Illumina Novaseq 6000 instruments.

ChIA-PET data were processed and analyzed using ChIA-PIPE algorithm[52], with the following steps: (1) Linker filtering and read alignment: paired-end tag (PET) read sequences were scanned for the bridge linker sequence and only PETs with the bridge linker were retained for downstream processing. After trimming the linkers, the flanking sequences were mapped to the human reference genome (hg38), and only uniquely-aligned PETs (MAPQ ≥ 30) were kept. These reads were used for PET clustering analysis after pruning duplicate reads; (2) PET clustering: The PET reads with a linker sequence detected with genomic tags at both ends were utilized to call interaction loops. These reads were categorized as either self-ligation PET (two ends of the same DNA fragment) or inter-ligation PET (two ends from two different DNA fragments in the same chromatin complex). Only clustered inter-ligation PETs with genomic span ≥8 kb reflect the long-range interactions of interest. Un-clustered inter-ligation PETs were considered as PET singletons. The number of PET read sequences (PET counts) represents the relative frequency of chromatin looping between two genomic loci; and we used a cutoff of PET count ≥2 to define high-confidence chromatin interactions. The final lists of chromatin interactions were provided as Supplementary Data 1 and 2 for ZNF384 ChIA-PET in TCZ PDX samples and JIH5 cells, respectively, and as Supplementary Data 3 and 4 for CTCF and RNAPII ChIA-PET in GM12878 (4D Nucleome Data portal accession number: 4DNEXKAMU4M6 and 4DNESFII9CQX). Finally, we also sought to explore protein binding sites across the genome (e.g., ZNF384 binding sites) using both inter-ligation and self-ligation reads from ChIA-PET; and binding peaks were called using MACS2[40]. MACS2 estimates a MAT score for each peak based on the signal to background ratio and a default *P* value cutoff of $1 \times 10^{-5}$ was used. Chromatin looping events are visualized using GenomePaint browser (https://viz.stjude.cloud/tools/genomepaint) with standard parameters[53].

## Ex vivo leukemia drug-sensitivity assay

Drug response of primary human ALL cells was evaluated using a co-culture system and high-content-imaging[54]. Leukemia samples were collected with informed consent as approved by St. Jude institutional review board. Demographic information (sex and age group) of ALL cases used this study is shown in Supplementary Table 3. hTERT-immortalized mesenchymal stem cells[55] (provided by Dr. Campana from National University of Singapore) were first seeded in a 384-well plate format at a density of 2500 cells per well in 25 μl of MSC medium 24 hours prior PDX sample preparation. After 24 hours, MSC medium was removed, and wells were washed with AIM-V medium (Gibco, #12055-083). Sorted leukemia cells were added at 25,000 cells per well to the stromal cell layer in 40 μl AIM-V medium along with 10 μl of drug solution prepared in the same medium. Triplications were included for each drug concentration/combination. After 96 hours of incubation at 37 °C with 5% $CO_2$, cells were harvested and stained with CyQUANT direct cell proliferation assay stain (Invitrogen, #C35011). After 15-30 minutes of incubation at 37 °C with 5% $CO_2$, the assay plate was placed in the high-content-imaging analysis system (PerkinElmer, Operetta CLS #HH16000000). Cell viability data were analyzed by Harmony high-content imaging and analysis software (version 4.9).

## Mouse studies

NOD.Cg*Prkdc*^scid^*Il2rg*^tm1Wjl^/SzJ (NSG) mice were purchased from Animal Resource Center at St. Jude Children's Research Hospital. All NSG mice were maintained in sterilized conditions with a temperature between 20 and 23 °C and humidity between 40 and 60% with 12-hour light/12-hour dark cycle. Animal experiments were performed according to procedures approved by the St. Jude Children's Research Hospital Institutional Animal Care and Use Committee. Two million primary ALL blasts or JIH5 cells were injected into five female NSG mice with ages between 8 and 12 weeks by tail vein injection. Treatment with gilter-itinib at 10 mg/kg was given by oral garage daily three days after transplantation, as described previously[54,56]. Starting from two weeks after injection, blast percentage in the peripheral blood was monitored by tracking the level of human leukemia cells using flow cytometry using BD FACS LSR II machine (BD FACS Diva Software V8.0) (cells were stained with human CD45-FITC [BD Pharmingen, #555482, Clone HI30, dilution ratio: 1:100], mouse CD45-APC-Cy7 [BD Pharmingen, #557659, Clone 30-F11, dilution ratio: 1:100], and mouse Ter119-PerCP-Cy5.5 [BD Pharmingen, #560512, Clone TER-119, dilution ratio: 1:100]). Mice were sacrificed when either blast percentage reached 80% in peripheral blood or mice reached the humane endpoint. Gating strategy for identifying human ALL blasts is shown in Supplementary Fig. 10.

## Enhancer analysis for the *FLT3* locus using ENCODE data

ENCODE H3K27ac ChIP-seq data from 22 cancer cell lines, normal human cells, or tissues (DND41, K562, A549, Hela, HepG2, HCT-116, MCF-7, PANC-1, CD20+, GM12878, CD14+, HMEC, NH-A, NHDF, NHEK, NHLF, Osteoblast, H1-hESC, HSMM, HSMMtube, HUVEC, and AG04450) were downloaded and aligned using Integrative Genomics Viewer (IGV)[57].

## Statistical analysis

All experiments were independently performed at least three times. The student's *t* test was used to calculate the *P* value for all analysis unless otherwise indicated. The correlation coefficient was calculated by Pearson's correlation. Graphpad Prism (version 9) was used to generate and expression plot, MTT plot, blast percentage plot, survival curve, and correlation plot. FlowJo software (Tree Star) was used to analyze flow cytometry results.

## Reporting summary

Further information on research design is available in the Nature Research Reporting Summary linked to this article.

# Data availability

All the sequencing data (CUT&RUN, ATAC-seq, and ChIA-PET) were uploaded to GEO (GSE197890). Previously published RNA-seq data can be available at EGAD00001004461, EGAD00001004463, EGAS00001001858, EGAS00001003726, and EGAS00001004532. The data generated in this work are provided in the Supplementary Information and Source Data files. Source data are provided with this paper.

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

## Acknowledgements

We thank the patients and parents who participated in the clinical trials included in this study. We thank Dr. Chunliang Li for his advices on genomic data analysis. We thank Dr. Guoqing Du for his assistance with mouse studies and the Pharmacotyping Resource for performing ex vivo ALL drug sensitivity tests. This work was supported by the National Institutes of Health (P50GM115279 [J.J.Y.], CA21765 [J.J.Y.], CA234490 [D.S.], HG009409 [C.W.]), and by the American Lebanese Syrian Associated Charities (ALSAC). The content is solely the responsibility of the authors and does not necessarily represent the official views of the U.S. National Institutes of Health.

## Author contributions

J.J.Y. is the principal investigator of this study, has full access to all the data in the study, and takes responsibility for the integrity of the data and the accuracy of the data analysis. X.Z., P.W. performed the experiments, W.Y., B.X., and Z.L. performed data analysis. J.J.Y. and X.Z. wrote the manuscript. J.Z., Z.L., J.D., B.S., N.R., S.Y., K.B., C.W., K.C., A.Y., M.K., C.H.P., and D.S. contributed reagents, materials, and/or data. J.J.Y., X.Z., P.W., W.Y., B.X., X. H., J.Y., and D.S. interpreted the data and the research findings. All the co-authors reviewed the manuscript.

## Competing interests

J.J.Y. receives a research grant from Takeda Pharmaceutical Company. Other authors declare no competing interests.
