## [Peer Review File · Nature Communications]

REVIEWER COMMENTS

Reviewer #1 (Remarks to the Author); expert in paediatric leukaemia:

In this interesting manuscript the authors identified FLT3 as a transcriptional target of ZNF384 fusions and suggest that *flt3* inhibition is of potential therapeutic value in these leukemias.

The novelty of this paper is the demonstration of the direct regulation of FLT3 expression by ZNF384 fusions.

As clearly discussed by the authors, the place of FLT3 inhibitors for treatment of leukemias with high expression of non-mutated FLT3 is debatable. Convincing activity of these inhibitors has been only shown in leukemias with activating mutations in FLT3 (internal tandem duplication of point mutations). Hence evidence by more extended preclinical experiments is needed to substantiate the claims by the authors that this subtype of higher risk ALL (non ZNF384-EP300 are high-risk leukemia 2021) is sensitive to FLT3 inhibitors

Comments:

1. Magnitude of increased expression of FLT3: FLT3 ligand is an important growth factor for B cells, thus the expression of FLT3 in most B cell precursor ALLs (Fig 1) is not surprising. Although ZNF384 fusion leukemias have a higher expression of *flt3* than, for example, ETV6-RUNX1 leukemias, the absolute difference is not that impressive. It bears no similarity to the difference of expression of, for example, genes that are amplified. This is a modest increase of expression.
2. Is there evidence for increased expression of FLT3 protein on the surface of these leukemias, compared with controls? This can simply be done by flow cytometry.
3. I wonder if the expression of *flt3* is related to the differentiation stage of leukemias. It is interesting that the two subtypes of B-ALLs with the highest expression of *flt3*, *znf384* and *kmt2ar*, are the most immature one – both are often CD10 negative or weak and co-express myeloid antigens
4. Figure 4A – Perhaps I do not understand the figure – is the percent viability of the *tcf3-znf384* cells is dramatically lower (less than 75%) in the absence of drug?
5. Figure 4A: More ZNF384 cell lines need to be tested
6. Figure 4B/C: what was the viability of the *znf384* fusion ALLs after the knockout of *znf384*? If most of the cells die/don't grow, then the sensitivity to FLT3 inhibitors is meaningless.
7. To definitely show the selective dependence of *znf384*rALLs on FLT3 expression one needs to complement the pharmacological experiments with genetic experiments with knockdown of FLT3. As knockout of FLT3 may be lethal for B cell precursors, regardless of their genetic subtypes, I recommend a knockdown approach either by using dCas9 or RNAi. Obviously, it is important to do these experiments in several genetic subtypes of ALLs as proper controls
8. 4E and G – I believe that the labels were mixed up – the current labeling suggests that *flt3* inhibition accelerate death.

9. The in-vivo preclinical experiments need to be expanded so that they may serve as a basis for a clinical trial or individual decisions to treat a patient with refractory ALL with a FLT3 inhibitor. Specifically, additional PDXs from ZNF384 patients should be used with a trial design mimicking a clinical trial. Namely many PDXs, each injected into two mice randomized to either vehicle or drug. Would be best to compare with a control group of, for example, ETV6-RUNX1 ALL.

Reviewer #2 (Remarks to the Author); expert in epigenetics:

In this interesting manuscript, Zhao et al report the finding that ZNF384-rearranged ALLs show overexpression of FLT3 via a de novo enhancer bound by the fusion protein. Cases with ZNF384 rearrangement show the highest levels of FLT3. Further epigenetic profiling identified open chromatin, active enhancer marks and ZNF384 binding to this de novo enhancer region located 25kb upstream of FLT3, which is otherwise closed in non-rearranged leukemias. Luciferase reporter assays showed increased transactivation activity driven by these enhancer region, and ChIA-PET analyses confirmed physical interaction with the FLT3 promoter. Interestingly, FLT3 inhibitors show good therapeutic activity in ZNF384 rearranged cells, whereas non-rearranged cells (or cells with CRISPR-induced deletion of the enhancer) are less sensitive to them. Finally, treatment of ZNF384-rearranged PDXs with the FLT3 inhibitor Gilteritinib in mice led to therapeutic activity and extension in survival. Although the bigger question of how the fusion proteins drive the establishment of this enhancer is not addressed, overall, this is a very nice study and I have only minor comments for authors

1. Authors show chromosomal interaction between the enhancer and the FLT3 promoter, but there is also chromosomal interaction with the neighboring gene PAN3. Even if upon CRISPR-induced deletion they don't observe differences in PAN3, it would be worthy to check the levels of PAN3 in all of the panels shown in Fig 1, to check whether it is overexpressed paralleling FLT3.

2. Even if the reporter assay already shows increased transactivation for the enhancer, this is only done in one orientation, and enhancers are known to drive transcriptional activation independently of orientation. Thus, it would be nice if authors could provide a more complete panel 2D with the results of the enhancer cloned in both + and - orientations.

3. Is this enhancer only active in ZNF384 rearranged leukemias? It would be relevant that authors looked at the ENCODE panel of cancer cell lines and tissues, to explore whether this region shows activity in any other FLT3-expressing cell type or not. Slightly related to this, would overexpression of a ZNF384 fusion protein lead to the establishment of this enhancer in leukemia cells (while wt ZNF384 on its own might not?) and what about in non-B leukemia cells? or in easy to use epithelial cells such as 293Ts?

4. When authors perform CRISPR-induced knockdown of the fusion protein (Fig 4B), authors fail to show the expression of the fusion protein itself (mRNA/protein). This should be shown. Also, authors show deletion of the fusion protein leads to reduced sensitivity to FLT3 inhibition, but failed to show whether deletion of the fusion protein resulted in delayed proliferation/cell cycle arrest/apoptosis by itself. This data should be shown.

5. The kinetics of both in vivo experiments are wildly different, with the PDX showing 2-fold increase in survival with Gilteritinib, whereas in the JH5 cells it seems it would "only" give a 2-3 week difference. Even more surprising is that the JH5 cells take ~16 weeks to kill the mice, while the PDX takes only about 5 weeks. I would usually expect a cell line to be more aggressive than a recently established PDX. Can authors comment on these differences?

6. Authors find FLT3 mutations are uncommon in ZNF384-rearranged cases, still, they find 5/66 mutations (line 193). Are these mutations found in any specific common subset of rearrangements?

Reviewer #3 (Remarks to the Author); expert in bioinformatics, ChIA-PET:

Genomic rearrangement is one of the main factors in occurrence of leukemia and a better understanding of genomic landscape in leukemia will help us to identify new therapeutic targets for leukemia. In this manuscript, Yang and coauthors used two published cohorts and found that FLT3 was overexpressed in ZNF384-rearranged ALL. Further experiments showed that there was a special enhancer 25kb upstream of FLT3 promoter probably bound by ZNF384 fusion proteins. Knockout of this enhancer had the inhibition effect on FLT3 expression. Such finding may lead to potential treatment of the corresponding groups of leukemia patients with ZNF384-rearranged ALL. The methods in this manuscript are straightforward and the finding may be beneficial to the sub-group of leukemia patients. Before it is considered for publication, a few issues need to be addressed first, which are listed as follows.

1. In Figure 1, the overexpression of FLT3 gene in ZNF384-rearranged ALL was shown. In Lines 262-263, it is mentioned that "FLT3 is rarely over-expressed in cases without ZNF384 fusion". So, it is better to add some negative controls without ZNF384 rearrangement in Figure 1 for comparison purpose.

2. ChIA-PET is a method to generate genome-wide chromatin interactions. It is better to summarize the entire libraries and demonstrate whether there are chromatin interactions from other genomic regions to FLT3 promoter besides those from the special enhancer to FLT3 promoter.

3. In Figure 3, the authors showed the chromatin interactions mediated by ZNF384 in TCZ cells. Have the authors generated ZNF384 ChIA-PET from JIH5 cell line? That should be a further support to the research.

4. One of the important experiments is CRISPR/Cas9 knockout of EP300-ZNF384 (EPZ) fusion gene in JIH5 cells. It is better to demonstrate in a figure or a supplemental figure to show the positions of the primers relative to the EP300-ZNF384 (EPZ) fusion gene.

Comments from the Reviewers

Reviewer #1 (Remarks to the Author): expert in paediatric leukaemia:

In this interesting manuscript the authors identified FLT3 as a transcriptional target of ZNF384 fusions and suggest that flt3 inhibition is of potential therapeutic value in these leukemias. The novelty of this paper is the demonstration of the direct regulation of FLT3 expression by ZNF384 fusions. As clearly discussed by the authors, the place of FLT3 inhibitors for treatment of leukemias with high expression of non-mutated FLT3 is debatable. Convincing activity of these inhibitors has been only shown in leukemias with activating mutations in FLT3 (internal tandem duplication of point mutations). Hence evidence by more extended preclinical experiments is needed to substantiate the claims by the authors that this subtype of higher risk ALL (non ZNF384-EP300 are high-risk leukemia 2021) is sensitive to FLT3 inhibitors.

Comments:

1. Magnitude of increased expression of FLT3: FLT3 ligand is an important growth factor for B cells, thus the expression of FLT3 in most B cell precursor ALLs (Fig 1) is not surprising. Although ZNF384 fusion leukemias have a higher expression of flt3 than, for example, ETV6-RUNX1 leukemias, the absolute difference is not that impressive. It bears no similarity to the difference of expression of, for example, genes that are amplified. This is a modest increase of expression.

The average *FLT3* expression is the highest in *ZNF384-r* ALL (14.12 +/- 0.87 in FPKM), compared to 11.63 (+/- 1.78) in other ALL subtypes, representing a 6.2-fold increase. In fact, this is higher than expression upregulation driven by chromosomal amplification in hyperdiploid ALL (e.g., average expression increase for genes on Chr21 is 1.5-fold in cases with somatic trisomy of Chr21 compared to those with diploid Chr21).

That said, we agree that the degree of upregulation seen with *FLT3* here is not as drastic as overexpression associated with enhancer hijacking events (e.g., *CRLF2* rearrangements in B-ALL or *TAL1* fusion in T-ALL). We reason that there is not necessarily a linear relationship between expression level and biological effects of a gene. As we have shown in a series of functional experiments, *FLT3* overexpression in *ZNF384-r* ALL is therapeutically important because FLT3 signaling is essential for leukemia survival in this context, even though the degree of overexpression is not the highest seen in this cancer.

2. Is there evidence for increased expression of FLT3 protein on the surface of these leukemias, compared with controls? This can simply be done by flow cytometry.

This is a great suggestion, and we compared the expression of FLT3 across a panel of ALL cell lines using flow cytometry. As shown below, FLT3 level on the cell surface was highest in *KMT2A-r* and *ZNF384-r* lines (SEM and JIH5) and was negatively correlated with gilteritinib LC50 across these models. We have now included these data in **Supplementary Fig. 4** (Panels **c** and **d** and see below).

3. I wonder if the expression of *flt3* is related to the differentiation stage of leukemias. It is interesting that the two subtypes of B-ALLs with the highest expression of *flt3*, *znf384* and *kmt2ar*, are the most immature one – both are often CD10 negative or weak and co-express myeloid antigens.

This is an excellent point and we have now explored it as followed. We first analyzed genome-wide gene expression data of normal hematopoietic cells (A comprehensive single cell transcriptional landscape of human hematopoietic cells [https://data.humancellatlas.org/explore/projects/5116c081-8be7-49c5-8ce0-73b887328aa9]) and identified 10 genes marking 7 stages of B cell differentiation (CLP, Pre-pro B, Pro-B, Cycling-Pre B, Pre-B, immature B, mature B). Then we examined the expression of these 10 genes across

416 ALL cases of five different subtypes (*ZNF384-r*, *KMT2A-r*, *ETV6-RUNX1*, *TCF3-PBX1*, and *MEF2D-r*), using the cohort1 dataset. We assigned a differentiation stage of each subtype and as shown below (also **Supplementary Fig. 8**), *FLT3* expression was highest in immature ALLs (*KMT2A-r* and *ZNF384-r*) and lower in those further differentiated (*ETV6-RUNX1*, *TCF3-PBX1*, and *MEF2D-r*).

We have now included these new results as **Supplementary Fig. 8** and also on **Page 17**.

4. Figure 4A – Perhaps I do not understand the figure – is the present

viability of the *tcf3-znf384* cells is dramatically lower (less than 75%) in the absence of drug?

We thank this comment. Actually, the viability was calculated by normalizing to no drug treatment control (as 100%).

5. Figure 4A: More *ZNF384* cell lines need to be tested

Unfortunately, JIH5 is the only available ALL cell line carrying the *ZNF384*-fusion gene across different repositories (e.g., ATCC and DSMZ).

As an alternative approach to address this, we have now tested a cohort of 47 primary ALL samples from patients for gilteritinib sensitivity, representing 7 subtypes including 3 cases with

ZNF384-r (see below and also **Fig. 4b**). Primary patient samples with *ZNF384*-fusion gene were highly sensitive to FLT3 inhibitor compared to other ALL subtypes.

6. Figure 4B/C: what was the viability of the *znf384* fusion ALLs after the knockout of *znf384*? If most of the cells die/don't grow, then the sensitivity to FLT3 inhibitors is meaningless.

This is a valid point and we agree. We have now analyzed cell cycle and viability of JIH5 cells after *EP300-ZNF384 (EPZ)* knockdown (using Cas9/CRISPR). Decreased expression of EPZ fusion was first confirmed at both mRNA and protein levels (**Fig. 4c and Supplementary Fig. 5a**).

There was no significant difference in cell cycle distribution between control cells (transduced with non-targeted sgRNA) and those with *EPZ* KD (**Supplementary Fig. 5c**). By contrast, downregulating *EPZ* expression led to a decrease in viability of JIH5 cells (**Supplementary Fig.**

5d). This is not surprising because EPZ fusion is likely a driver genomic abnormality for this subtype of ALL. We have added a new **Supplementary Fig. 5** to include these results.

It should be noted that this effect was already controlled for in the FLT3 inhibitor sensitivity analysis because viability was normalized against cells with EPZ knockdown but no drug treatment.

7. To definitely show the selective dependence of znf384rALLs on FLT3 expression one needs to complement the pharmacological experiments with genetic experiments with knockdown of FLT3. As knockout of FLT3 may be lethal for B cell precursors, regardless of their genetic subtypes, I recommend a knockdown approach either by using dCas9 or RNAi. Obviously, it is important to do these experiments in several genetic subtypes of ALLs as proper controls.

We appreciate this comment and have now performed the suggested experiments. We directly knocked down *FLT3* in three ALL cell lines (JIH5 [*EP300-ZNF384*], 697 [*TCF3-PBX1*] and REH [*ETV6-RUNX1*], see below). As shown by the flow cytometry, *FLT3* expression was efficiently suppressed in all three cell lines. However, only JIH5 cells showed decreased viability upon FLT3

knockdown, suggesting its essential role in *ZNF384-r* ALL. We have added a new **Supplementary Fig. 6** to include these results.

8. 4E and G – I believe that the labels were mixed up – the current labeling suggests that flt3 inhibition accelerate death.

We apologize and have now corrected the labels in **Fig. 4E and G**.

9. The in-vivo preclinical experiments need to be expanded so that they may serve as a basis for a clinical trial or individual decisions to treat a patient with refractory ALL with a FLT3 inhibitor. Specifically, additional PDXs from ZNF384 patients should be used with a trial design mimicking a clinical trial. Namely many PDXs, each injected into two mice randomized to either vehicle or drug. Would be best to compare with a control group of, for example, ETV6-RUNX1 ALL. We thank this reviewer for his/her comment which we agree. Unfortunately, it has proven very challenging to develop xenografts of *ZNF384-r* B-ALL which is a relatively rare subtype to start with. We injected 5 cases of primary ALL with this type of fusion, and only two engrafted, one of which exhibited low blast % even after 6 months and thus cannot be used for the efficacy study.

That said, we agree with this reviewer and certainly recognize the value of expanding the *ZNF384-r* ALL cohort. Therefore, to further confirm the therapeutic effects of gilteritinib on this subtype, we performed *ex vivo* drug-sensitivity assay in a panel of 47 primary ALL cases including three with *EP300-ZNF384* fusion as well as those with *ETV6-RUNX1*, *DUX4-r*, *BCR-ABL1*, hyperdiploidy, and T-ALL (see below, **also Fig. 4b**). Primary ALL samples with *ZNF384*-fusion gene showed superior sensitivity than those of other genotypes.

Reviewer #2 (Remarks to the Author); expert in epigenetics:

In this interesting manuscript, Zhao et al report the finding that ZNF384-rearranged ALLs show overexpression of FLT3 via a de novo enhancer bound by the fusion protein. Cases with ZNF384 rearrangement show the highest levels of FLT3. Further epigenetic profiling identified open chromatin, active enhancer marks and ZNF384 binding to this de novo enhancer region located 25kb upstream of FLT3, which is otherwise closed in non-rearranged leukemias. Luciferase reporter assays showed increased transactivation activity driven by these enhancer region, and ChIA-PET analyses confirmed physical interaction with the FLT3 promoter. Interestingly, FLT3 inhibitors show good therapeutic activity in ZNF384 rearranged cells, whereas non-rearranged cells (or cells with CRISPR-induced deletion of the enhancer) are less sensitive to them. Finally, treatment of ZNF384-rearranged PDXs with the FLT3 inhibitor Gilteritinib in mice led to therapeutic activity and extension in survival. Although the bigger question of how the fusion proteins drive the establishment of this enhancer is not addressed, overall, this is a very nice study and I have only minor comments for authors

1. Authors show chromosomal interaction between the enhancer and the FLT3 promoter, but there is also chromosomal interaction with the neighboring gene PAN3. Even if upon CRISPR-induced deletion they don't observe differences in PAN3, it would be worthy to check the levels of PAN3 in all of the panels shown in Fig 1, to check whether it is overexpressed paralleling FLT3.

As suggested, we have now examined *PAN3* expression in the same ALL cohorts used for the *FLT3* expression analyses. As shown in **Supplementary Fig. 7b and 7c**, *PAN3* was ubiquitously expressed across ALL subtypes. We included a brief note in the discussion to describe this (Page 17).

2. Even if the reporter assay already shows increased transactivation for the enhancer, this is only done in one orientation, and enhancers are known to drive transcriptional activation independently of orientation. Thus, it would be nice if authors could provide a more complete panel 2D with the results of the enhancer cloned in both + and - orientations.

We appreciate this helpful comment. We have generated a luciferase reporter vector with reversed z-FLT3 enhancer and performed the luciferase reporter assay in JIH5 cells. There was no significant difference between + and – orientations. We have now included this result in **Fig. 2D**.

3. Is this enhancer only active in ZNF384 rearranged leukemias? It would be relevant that authors looked at the ENCODE panel of cancer cell lines and tissues, to explore whether this region

shows activity in any other FLT3-expressing cell type or not. Slightly related to this, would overexpression of a ZNF384 fusion protein lead to the establishment of this enhancer in leukemia cells (while wt ZNF384 on its own might not?)? and what about in non-B leukemia cells? or in easy to use epithelial cells such as 293Ts?

We appreciate these helpful comments.

1) We have explored all the available H3K27ac ChIP-seq data of a series of 8 cancer cell lines

and 14 normal human cells/tissues included in ENCODE (see below, **Supplementary Fig. 3**). The z-FLT3 enhancer (upper panel, ATAC-seq data from PDX EPZ) was observed in none of them, suggesting that this enhancer is unique to *ZNF384-r* ALL.

2) As suggested, we have now overexpressed the EP300-ZNF384 (EPZ) fusion gene in 697, REH, and 293T cells. *FLT3* expression was determined using flow cytometry (see below). *FLT3* level

increased modestly as a result of ectopic expression of the EPZ fusion gene in ALL cell lines, but this was not true in 293T cells.

4. When authors perform CRISPR-induced knockdown of the fusion protein (Fig 4B), authors fail

to show the expression of the fusion protein itself (mRNA/protein). This should be shown. Also, authors show deletion of the fusion protein leads to reduced sensitivity to FLT3 inhibition, but failed to show whether deletion of the fusion protein resulted in delayed proliferation/cell cycle arrest/apoptosis by itself. This data should be shown.

We agree and have now quantified EPZ fusion protein (normalized to actin) in JIH5 cells and confirmed the efficiency of EPZ knockdown (63% of control).

As also suggested by Reviewer 1, we now checked the effects of EPZ knockdown on ALL cell proliferation, apoptosis, and cell cycling. There was no significant difference in cell cycle distribution between EPZ knockdown (KD) cells and control cells. However, the viability of EPZ KD cells was indeed lower than control cells (transduced with non-targeted sgRNA).

5. The kinetics of both *in vivo* experiments are wildly different, with the PDX showing 2-fold increase in survival with Gilteritinib, whereas in the JH5 cells it seems it would "only" give a 2-3 week difference. Even more surprising is that the JH5 cells take ~16 weeks to kill the mice, while the PDX takes only about 5 weeks. I would usually expect a cell line to be more aggressive than a recently established PDX. Can authors comment on these differences?

This reviewer is correct that our two *ZNF384-r* ALL models have different growth rate *in vivo*. This is not unusual as the cellular features of different cases can vary significantly despite having the same sentinel fusion gene (i.e., *ZNF384*-fusion). We have examined the PROPEL portal with 66 ALL PDX models (<https://propel.stjude.cloud/>) and plotted the time to engraftment of cases carrying *BCR-ABL1*, *ETV6-RUNX1*, and *KMT2A-r*. As seen below, there were wide variations within the same ALL subtype in terms of engraftment (and potentially growth *in vivo*). JH5 cell line is actually very slow-growing *in vitro* compared to other ALL cell line models, with a doubling time of 5 days. The *in vivo* growth kinetics of leukemia cells can be influenced by a range of factors, e.g., efficiency in bone marrow homing and engraftment, proliferation rate *in vivo*, extramedullary disease presentation and in the peripheral blood.

We added discussion of this issue on Page 15.

6. Authors find FLT3 mutations are uncommon in *ZNF384*-rearranged cases, still, they find 5/66 mutations (line 193). Are these mutations found in any specific common subset of rearrangements?

As suggested, we have updated **Fig. 1** to indicate cases with *FLT3* mutation within *ZNF384-r* ALL (green triangle). We also now describe the exact *FLT3* mutations in *ZNF384-r* ALL in these two cohorts, as shown in the Protein paint (see below and **Fig. 1e**).

Figure 1

Reviewer #3 (Remarks to the Author); expert in bioinformatics, ChIA-PET:

Genomic rearrangement is one of the main factors in occurrence of leukemia and a better understanding of genomic landscape in leukemia will help us to identify new therapeutic targets for leukemia. In this manuscript, Yang and coauthors used two published cohorts and found that FLT3 was overexpressed in ZNF384-rearranged ALL. Further experiments showed that there was a special enhancer 25kb upstream of FLT3 promoter probably bound by ZNF384 fusion proteins. Knockout of this enhancer had the inhibition effect on FLT3 expression. Such finding may lead to potential treatment of the corresponding groups of leukemia patients with ZNF384-rearranged ALL. The methods in this manuscript are straightforward and the finding may be beneficial to the sub-group of leukemia patients. Before it is considered for publication, a few issues need to be addressed first, which are listed as follows.

1. In Figure 1, the overexpression of FLT3 gene in ZNF384-rearranged ALL was shown. In Lines 262-263, it is mentioned that "FLT3 is rarely over-expressed in cases without ZNF384 fusion". So, it is better to add some negative controls without ZNF384 rearrangement in Figure 1 for comparison purpose.

We appreciate this comment. In **Fig. 1**, ALL cases were classified into 23 mutually exclusive subtypes. Therefore, if a case is labeled as anything other than *ZNF384-r*, it is negative for this fusion. 1,939 and 360 cases in Cohort 1 and 2 were indeed without *ZNF384* fusion and are considered as negative control for comparison purpose.

In contrast to high expression of *FLT3* in *ZNF384*-rearrangement, *KMT2A*-rearrangement, and hyperdiploidy subtypes, low expression of *FLT3* was observed in *TCF3-PBX1*, *NUTM1*, and *BCL2/MYC* subtypes.

2. ChIA-PET is a method to generate genome-wide chromatin interactions. It is better to summarize the entire libraries and demonstrate whether there are chromatin interactions from other genomic regions to FLT3 promoter besides those from the special enhancer to FLT3 promoter.

We agree completely and have now provided the complete summary of chromatin interactions identified in these genome-wide assays in TCZ PDX and JH5 cells (as **Supplementary Table 3 and 4**).

3. In Figure 3, the authors showed the chromatin interactions mediated by ZNF384 in TCZ cells. Have the authors generated ZNF384 ChIA-PET from JIH5 cell line? That should be a further support to the research.

We agree and have now performed ZNF384 ChIA-PET in JIH5 cells. As shown below, we observed similar pattern of chromatin interactions at the *FLT3* locus, consistent with ZNF384 ChIA-PET results in TCZ PDX cells. We updated **Fig. 3** to include the new results.

4. One of the important experiments is CRISPR/Cas9 knockout of EP300-ZNF384 (EPZ) fusion gene in JH5 cells. It is better to demonstrate in a figure or a supplemental figure to show the positions of the primers relative to the EP300-ZNF384 (EPZ) fusion gene.

As suggested, we have added a new **Supplementary Fig. 1** to show the positions of primers and sgRNA targeting the EPZ fusion gene.

REVIEWER COMMENTS

Reviewer #1 (Remarks to the Author):

Thank you for a comprehensive response to my inquiries. I am convinced now with both the novelty and, more importantly, of the potential translational aspects of this study. This paper may convince pharma to expand the pediatric phase II trials with Gliterinib to these rare patients.

Reviewer #2 (Remarks to the Author):

Authors did a good job of answering my comments.

Still, I was asked to specifically comment on their responses to original Reviewer#3, and I would like some further clarification in their response to Reviewer#3, point #2. While authors now provide a complete summary of chromatin interactions identified in the ChIA-PET assays in TCZ PDX and JIH5 cells in Supplementary Table 3 and 4, as requested by the reviewer, I have some questions about this:

1. Supplementary Tables 3 and 4 are not mentioned in the manuscript. Authors should of course mention them in the main text.
2. The titles of both Supplementary Tables 3 and 4 are the same (Genome-wide DNA interaction mediated by ZNF384 in TCZ PDX sample), even if the interactions listed are different. Please revise so that it's clear which table corresponds to TCZ cells and which table corresponds to JIH5 cells.
3. For comparison purposes, and because authors do have these data already, the ChIA-PET signal from their GM12878 (non-ZNFrearranged) controls should also be shown in another supplementary table with the complete summary of interactions.
4. Finally (and maybe the most relevant comment, as the comments above are really minor), looking at the excel table as currently presented, the ZNF-mediated interactions between the enhancer and promoter are not entirely clear to me. In line with this, Fig 3 shows several looping interactions (the red curves), but it is not described anywhere what are the thresholds used to show those specific interactions/red curves and not others. This is relevant given the huge amount of data obtained in the ChIA-PET. How did authors select those specific interactions? Maybe using different colors to directly highlight in the excel Supp Tables which interactions are shown by the red curves in Fig 3 would be a helpful thing to add. Overall, I would expect authors to revise this so that they can clearly show that the interactions between enhancer-promoter are not just random, but more frequent/significant than interactions observed between the promoter and other regions, or between regions not considered either as enhancers or promoters.

Comments from the Reviewers

Reviewer #1 (Remarks to the Author):

Thank you for a comprehensive response to my inquiries. I am convinced now with both the novelty and, more importantly, of the potential translational aspects of this study. This paper may convince pharma to expand the pediatric phase II trials with Gliterinib to these rare patients.

Thank you.

Reviewer #2 (Remarks to the Author):

Authors did a good job of answering my comments.

Still, I was asked to specifically comment on their responses to original Reviewer#3, and I would like some further clarification in their response to Reviewer#3, point #2. While authors now provide a complete summary of chromatin interactions identified in the ChIA-PET assays in TCZ PDX and JIH5 cells in Supplementary Table 3 and 4, as requested by the reviewer, I have some questions about this:

1. Supplementary Tables 3 and 4 are not mentioned in the manuscript. Authors should of course mention them in the main text.

We agree and have updated the main text accordingly. These Supplementary Tables are now cited on **Page 14**.

2. The titles of both Supplementary Tables 3 and 4 are the same (Genome-wide DNA interaction mediated by ZNF384 in TCZ PDX sample), even if the interactions listed are different. Please revise so that it's clear which table corresponds to TCZ cells and which table corresponds to JIH5 cells.

We apologize for the confusion and have now revised the title of these two tables: **Supplementary Table 3: Genome-wide DNA interaction mediated by ZNF384 in TCZ PDX samples; Supplementary Table 4: Genome-wide DNA interaction mediated by ZNF384 in JIH5 cells.**

3. For comparison purposes, and because authors do have these data already, the ChIA-PET signal from their GM12878 (non-ZNF384 rearranged) controls should also be shown in another supplementary table with the complete summary of interactions.

We thank this comment and have now summarized the interactions mediated by CTCF and RNAPII in GM12878 cells as well using publicly available data (4D Nucleome Data Portal accession number: 4DNEXKAMU4M6 and 4DNESFII9CQX). We have now included these results in the new **Supplementary Tables 5 and 6** and they were cited on **Page 9**.

4. Finally (and maybe the most relevant comment, as the comments above are really minor), looking at the excel table as currently presented, the ZNF-mediated interactions between the enhancer and promoter are not entirely clear to me. In line with this, Fig 3 shows several looping interactions (the red curves), but it is not described anywhere what are the thresholds used to show those specific interactions/red curves and not others. This is relevant given the huge amount of data obtained in the ChIA-PET. How did authors select those specific interactions?

We appreciate these comments. We have added a detailed description of ChIA-PET data analysis in the Methods and Materials section, especially regarding how interactions were called by ChIA-PIPE software and the cutoffs used to distinguish signals from the background (**Pages 9 and 10**). Importantly, of chromatin looping called by ChIA-PIPE, we only kept those with at least 2 paired-

end tag count because these events were likely of high confidence (paired-end tag count represents the frequency of a given looping event). These analytical procedures are well-established and extensively described in our prior publications (*Curr Protoc.* 2021, 1(8):e174; *Genome Biol.* 2020 21(1):110; *Cell* 2015, 163(7):1611-27; *Nat Protoc.* 2017, 12(5):899-915).

We also updated **Figure 3** accordingly with details of how these chromatin loops were selected for illustration. Briefly, the red arches shown in **Figure 3** were chromatin interactions (loops) with paired-end tag (PET) counts ≥ 2 , which is the same as those in **Supplementary Tables 3-6**. No random chromatin interactions were shown in **Figure 3** or included in **Supplementary Tables 3-6** because they were already removed using the ChIA-PIPE pipeline (see Methods and Materials for details).

Our ChIA-PET experiment broadly detects ZNF384-mediated chromatin interactions in a genome-wide fashion. At the *FLT3* locus, some of these interactions are anchored at the intronic enhancer (indicated by the green arrow in **Figure 3**), including the enhancer-promoter looping as this reviewer alluded. However, we also detected chromatin looping events that do not directly involve this intronic enhancer but may still contribute to the overall 3D chromatin structure at this locus. Both types of looping are presented in **Figure 3** and **Supplementary Tables 3 and 4** so that the readers can evaluate the full extent of ZNF384-related interactions.

Maybe using different colors to directly highlight in the excel Supp Tables which interactions are shown by the red curves in Fig 3 would be a helpful thing to add.

This is a great suggestion. We have now updated **Supplementary Tables 3 and 4** by highlighting those ZNF384-mediated interactions shown by the red curves in **Figure 3**.

Overall, I would expect authors to revise this so that they can clearly show that the interactions between enhancer-promoter are not just random, but more frequent/significant than interactions observed between the promoter and other regions, or between regions not considered either as enhancers or promoters.

REVIEWERS' COMMENTS

Reviewer #2 (Remarks to the Author):

Authors did a good job of answering my last comments.

I would only suggest that they mention somewhere (Fig 3 legend? Supplementary Tables legend?) that the interactions highlighted in red in the Supp Tables correspond to the loopings shown in Fig 3.

Other than that, I would like to congratulate them on their very nice story.

Response to Reviewers

Comments from the Reviewers

Reviewer #2 (Remarks to the Author):

Authors did a good job of answering my last comments. I would only suggest that they mention somewhere (Fig 3 legend? Supplementary Tables legend?) that the interactions highlighted in red in the Supp Tables correspond to the loopings shown in Fig 3.

Other than that, I would like to congratulate them on their very nice story.

We appreciate this comment. We have now added a footnote for Supplementary Tables 1 (Genome-wide DNA interaction mediated by ZNF384 in TCZ PDX samples) and 2 (Genome-wide DNA interaction mediated by ZNF384 in JIH5 cells) to describe the rows highlighted in red.